# Pre-diagnostic trajectories of lymphocytosis predict time to treatment and death in patients with chronic lymphocytic leukemia

Michael Asger Andersen[1,2], Mia Klinten Grand [3], Christian Brieghel [1], Volkert Siersma[3], Christen Lykkegaard Andersen[1,3] & Carsten Utoft Niemann [1,4 ✉]

## Abstract

**Background** The dynamics of pre-diagnostic lymphocytosis in patients with ensuing chronic lymphocytic leukemia (CLL) need to be explored as a better understanding of disease progression may improve treatment options and even lead to disease avoidance approaches. Our aim was to investigate the development of lymphocytosis prior to diagnosis in a population-based cohort of patients with CLL and to assess the prognostic information in these pre-diagnostic measurements.

**Methods** All patients diagnosed with CLL in the Greater Copenhagen area between 2008 and 2016 were included in the study. Pre-diagnostic blood test results were obtained from the Copenhagen Primary Care Laboratory Database encompassing all blood tests requested by Copenhagen general practitioners. Using pre-diagnostic measurements, we developed a model to assess the prognosis following diagnosis. Our model accounts for known prognostic factors and corresponds to lymphocyte dynamics after diagnosis.

**Results** We explore trajectories of lymphocytosis, associated with known recurrent mutations. We show that the pre-diagnostic trajectories are an independent predictor of time to treatment. The implementation of pre-diagnostic lymphocytosis slope groups improved the model predictions (compared to CLL-IPI alone) for treatment throughout the period. The model can manage the heterogeneous data that are to be expected from the real-world setting and adds further prognostic information.

**Conclusions** Our findings further knowledge of the development of CLL and may eventually make prophylactic measures possible.

### Plain language summary

While clinicians largely agree that patients with chronic lymphocytic leukemia (CLL) have increased levels of white blood cells in the years preceding their diagnosis, there is less certainty as to how and when this increase occurs. A better understanding of how white blood cell levels change during this period might help us to predict who will become ill and require treatment. In this work, we explore patterns of white blood cell growth and develop a tool to predict the time to treatment for CLL based on these growth rates. Using our tool in the clinic might help clinicians to decide who needs treatment for CLL and when, potentially leading to better outcomes for patients.

[1] Department of Hematology, Rigshospitalet, Copenhagen University Hospital, Copenhagen, Denmark. [2] Department of Clinical Pharmacology, Bispebjerg Hospital, Copenhagen, Denmark. [3] The Research Unit for General Practice and Section of General Practice, Department of Public Health, University of Copenhagen, Copenhagen, Denmark. [4] Institute for Clinical Medicine, Copenhagen University, Copenhagen, Denmark. ✉email: carsten.utoft.niemann@regionh.dk

Chronic lymphocytic leukemia (CLL) is characterized by an accumulation of clonal, mature, CD5+ B cells in the peripheral blood, bone marrow, and secondary lymphoid organs[1]. The presence of more than 5000 clonal B cells per μL in peripheral blood is diagnostic for CLL[1, 2]. Prior to the diagnosis, clonal B cells develop from undetectable levels through (undetected) monoclonal B cell lymphocytosis (MBL) to overt lymphocytosis and finally to a state that exceeds the diagnostic criterion for CLL[3]. Consequently, lymphocytosis has often been present years prior to diagnosis, and at diagnosis, the clinicians will occasionally be able to assess this accumulation of the clone in terms of lymphocytosis in blood samples taken years before diagnosis.

Once diagnosed, the clinical course of CLL is very heterogeneous with some patients living for decades without needing treatment, while others have short survival despite multiple lines of intensive treatments[4, 5]. The diagnosis of CLL is often incidental, only 5% of all patients require treatment at the time of diagnosis[5] while the remaining 95% of patients are observed until signs of active disease. Roughly estimated, more than 400 000 individuals with CLL are in observational "watch and wait" for CLL in Europe and the United States[6, 7]. The current and most widely used prognostic model, the international prognostic index (CLL-IPI), is based on five variables—age, stage, TP53 aberration, IGHV mutational status, and beta-2-microglobulin[8]. Unfortunately, CLL-IPI is poor to predict disease progression in patients with asymptomatic early-stage CLL[9].

One important risk factor for almost all cancers is the rate by which it grows; the same is true for CLL. Previous studies have reported that the cancer growth rate carries prognostic information and progressive lymphocytosis with an increase of 50% over a 2-month period or lymphocyte doubling time <6 months are indications for treatment[2, 10, 11]. Cancer growth is, however, difficult to assess because CLL cells are located and proliferate in three different compartments—peripheral blood, lymph nodes, and bone marrow[10]. On one hand, the absolute lymphocyte count (ALC) in peripheral blood is easy to measure, but the proliferation rate for circulating CLL cells is only between 0.1 and 1% per day, not even taking into account the proportion of CLL cells disappearing due to apoptosis and necrosis[10]. Nevertheless, Gruber showed that genomic features correlate to individual patterns of lymphocyte kinetics in peripheral blood after diagnosis of CLL[12]. Thus, the next step toward understanding this heterogonous cancer would be to characterize the clonal development prior to the diagnosis of CLL. Such knowledge can potentially help define a more clinical meaningful transformation point from monoclonal B-lymphocytosis to CLL than the current somehow arbitrary 5000 clonal B cells per μL. Moreover, trajectories of important biomarkers prior to diagnosis of CLL may carry important prognostic information.

Thus, we investigated 10-year trajectories of hemoglobin, C-reactive protein (CRP), ALC, and platelet count prior to diagnosis of CLL based on a matched case-control population, with 15 random controls (CLL-free individuals) per case (CLL patients). We further assessed, compared, and validated the distribution of genetic features in a subpopulation with the observations from Gruber and colleagues on the correlation between post-diagnostic lymphocyte kinetics and genetic aberrations in patients with CLL[12]. Lastly, we developed a model based on the trajectories to stratify individuals at high risk of treatment following the CLL diagnosis.

## Methods
### Study population
*Retrospective CLL cohort.* We included all patients diagnosed with CLL in Greater Copenhagen between 2008 and 2016 through the Danish CLL Register (DCLLR). Since 2008, all newly diagnosed CLL patients are registered in the DCLLR[13]. All hematological departments in Denmark are required to report clinical data for patients with CLL and the data completeness in the DCLLR is 98.3%[13]. The DCLLR holds information on birthdays, treating hospitals, prognostic markers, and treatment initiations. From this database, we obtained information regarding age at diagnosis, Binet stage, β-2-microglobulin, fluorescence in situ hybridization data, IGHV mutational status, and date of first-line treatment or death (if applicable). CLL-IPI was calculated using B2M cutoff of 4 and del(17p) as the only TP53 aberration[8].

*Matched case–control population.* Each CLL patient with at least one measurement within 5 years of diagnosis of hemoglobin, CRP, ALC, or platelet count available were matched on sex and birth year with 15 random controls per measurement. Controls were free from CLL at the index date. Vital information on controls was obtained using the Danish Civil Registration System. The Danish Civil Registration system is for practical purposes complete and no persons are lost to follow-up[14].

**Variables.** The biochemistry data used in this study come from two different data sources. 1) The Centre of Excellence for Personalized Medicine of Infectious Complications in Immune Deficiencies (PERSIMUNE) and 2) The Copenhagen Primary Care Laboratory (CopLab) database.

PERSIMUNE contains data from all hospitals in Denmark. PERSIMUNE has streamlined and quality-checked the data received from the many different laboratories within Denmark.

The Copenhagen Primary Care Laboratory (CopLab) database. In the Copenhagen area (the Copenhagen Community and the former Copenhagen County), with its approximately 1.2 million inhabitants, there was only one laboratory serving general practitioners and practicing specialists until 2015, the Copenhagen General Practitioners' Laboratory (CGPL). The CGPL was accredited for International Organization for Standardization (ISO) standards ISO17025 and ISO15189 and has saved all data on the analyses it has performed since July 2000. The Copenhagen Primary Care Laboratory (CopLab) database contains all results ($n = 178,000,000$) from 1 July 2000 to 31 December 2015 from the CGPL[15].

Analytical methods of the ALC, platelet count, and hemoglobin are described elsewhere[16]. CRP may correlate with lymphocyte/leukocyte changes due to infection and has been predictive in many cancers[17]. CRP was assessed in serum by commercially available assays from Olympus A600 (Olympus A/S, Denmark) and ADVIA Chemistry System (Bayer®/Siemens®, Denmark) according to the instructions of the manufacturers. Two ADVIA Chemistry assays were used, including the C-reactive protein (CRP). For the Olympus assay, the inter-serial coefficient of variation percentage was 5.1% (at level 7 mg/L), 1.2% (at level 19 mg/L) and 2.5% (at level 31.5 mg/L). For the Advia Chemistry CRP assay, the inter-serial coefficients of variation percentage was 12.7% (at level 12 mg/L), 4.0% (at level 24.7 mg/L) and 2.6% (at level 98.6 mg/L). For the Advia Chemistry CRP_2 assay, the inter-serial coefficients of variation percentage was 1.59% (at level 23.5 mg/L), 1.45% (at level 75.5 mg/L) and 2.6% (at level 86.1 mg/L). The CRP assays were subject to external quality control through participation in the LabQuality External Quality Assessment Scheme (LabQuality, Helsinki, Finland). The assessment scheme included 12 distributions (each distribution comprised 1 sample) or 4 distributions (each distribution comprised 2 samples). The results through the entire CopLab period (from 2001 to 2015) confirmed the reliability of the assays, and the results from CGPL deviated <1.3% from the method mean. Low results were reported as <3 mg/L (until December 1,

2002), <5 mg/L (between December 2, 2002 and May 28, 2008) and <4 (from May 29, 2008). High results were reported as >300 mg/L (until December 1, 2002).

**Sequencing and bioinformatic analysis**. Next-generation sequencing has been described elsewhere[18]. Twenty-five driver mutations were assigned to signaling pathways according to consensus among two seminal papers[19, 20], and a pathway was considered altered if at least one gene in the pathway was mutated. Mutated genes involved in multiple pathways count as multiple altered pathways.

**Statistics and reproducibility**

*ALC, platelet count, CRP, and hemoglobin*. The matched population was used to estimate population-averaged trajectories separately for each biomarker. The marginal models for lymphocytosis, hemoglobin, and platelet counts were fitted using generalized estimating equations. A Gaussian parametric survival model, with a robust variance estimator, was used to account for the left censoring of CRP values below the detection limit. ALC, CRP, and platelet counts were log-transformed. We included time to diagnosis, classification group (control or case), sex, and age at diagnosis as covariates, along with an interaction between time to diagnosis and classification group. Higher-order polynomials of time to diagnosis were included, if significant, to allow for non-linear trajectories.

*Prognostic significance of lymphocyte slope*. We fitted a linear mixed model with random intercept and slope to define pre-diagnostic lymphocyte slope groups. We used the ALCs, from three years prior to diagnosis, from the CLL cohort patients. ALC was log-transformed and time to diagnosis was included as a covariate. The three slope groups (low, medium, high) were defined as the tertiles of the estimated random slopes. We then fitted Cox proportional hazard models for the cause-specific hazards of death and treatment. The timescale was time from diagnosis until the first event of death or treatment. The models were adjusted for slope group, sex, and CLL-IPI. We performed model control on all models. We compared the difference in AUC over time between the models including and not including the slope group to assess the predictive performance of the slope group[21]. The study was approved by the Danish National Committee on Heath Research Ethics and informed consent was not required for retrospectively included patients according to Danish legislation.

**Reporting summary**. Further information on research design is available in the Nature Research Reporting Summary linked to this article.

## Results

**Overall**. In total, we identified 1123 CLL patients diagnosed between 2008 and 2016 in the Capital Region of Denmark, with a median follow-up of 2.9 years before diagnosis and 2.3 years after diagnosis (Fig. 1). We included 4073 pre-diagnosis and 16,958 post-diagnosis ALC measurements. In the matched case-control population, including only patients with at least one ALC measurement prior to diagnosis in the CopLab laboratory database, we identified 778 CLL patients, who were matched with 15,124 controls free from CLL (Table 1). The median time from first ALC measurement to diagnosis of CLL was 7.8 years (interquartile range [IQR] 4.5–10.8). The median age at the date of diagnosis was 69 years (IQR 62–77).

**ALC, platelet count, CRP, and hemoglobin**. At the time of diagnosis (year 0), patients with CLL had a mean estimated ALC of 15,000 per μL compared with 2400 per μL for controls in accordance with the criteria for CLL diagnosis requiring at least 5000 clonal lymphocytes per μL. The gradual increase in ALC for CLL patients could be detected three to seven years before the diagnosis of CLL, whereas the ALC for controls was constant. For CRP and hemoglobin concentration, no differences were observed between cases and controls up until 1 year before the diagnosis of CLL, with both hemoglobin and CRP declining with increasing age. However, mean platelet count was inversely correlated with the mean ALC (Fig. 2 and Supplementary Figs. 1 and 2).

**Prognostic significance of ALCs**. Following this observation, we evenly divided patients in three slope groups based on their pre-diagnostic ALC growth rate; low slope ($n = 292$), medium slope ($n = 291$), and high slope ($n = 292$). We excluded 248 patients who did not have at least one observation within 5 years of diagnosis. Low slope groups had a growth rate of $<0.32 \times 10^9$ lymphocytes/L/per year, compared with the high slope group with a growth rate of $>0.44 \times 10^9$ lymphocytes/L/per year.

Adjusted for CLL-IPI and sex, the pre-diagnostic high slope group was an independent predictor for time to treatment compared with the low slope group (hazard ratio [HR] 2.44; 95% CI [1.69, 3.54]; $p < 0.001$). The time to treatment was not different between low and intermediate slope groups. Neither was the patient-specific pre-diagnostic lymphocyte kinetics significantly associated with the risk of death for high slope compared to the low slope (HR 1.64; 95% CI [0.97, 2.76]; $p = 0.06$; Figs. 2 and 3 and Supplementary Tables 1 and 2). The estimated 1-year cumulative incidences for treatment, for a male with CLL-IPI 1 were 15.2, 6.1, and 6.6% for the high slope group, medium slope group, and low slope group. The addition of the slope group improved the model predictions (compared to CLL-IPI alone) for both treatment and death throughout the period, e.g., at 1 year after diagnosis the AUC was improved by 2.2 percentage points (95% CI [−1.5, 6.0]) for death and with 4.5 percentage points (95% CI [0.2, 8.9]) for treatment. However, as expected, the predictive effect of the pre-diagnostic ALC slope group declined over time (Supplementary Figs. 3).

**Growth patterns influence growth dynamics after diagnosis**. Next, we used the Bayesian Network, developed by Gruber and colleagues[12], to classify the growth rates of ALC kinetics after CLL diagnosis. We found that 246 patients exhibited logistic (LOG) growth behavior. In 25 patients, the CLL cells exhibited exponential (EXP) growth, and 277 patients belonged to the indeterminate (IND) group either because of the short period of observation or owing to complex patterns of growth. The unknown group (569 patients) had less than three observations after diagnosis, which was required for the Bayesian Network to analyze growth kinetics. For six patients, the Bayesian Network did not yield any results (Fig. 4).

Next, we compared the pre-diagnostic slope groups to the post-diagnostic Bayesian Network. In general, the pre-diagnostic slope groups were associated with the Bayesian Network post-diagnostic growth patterns (chi-square $p$ value 0.02). Thus, we found a higher proportion of patients with EXP growth among patients with high and medium pre-diagnostic slope (5.3 and 5.7%) compared with the low slope group (3.3%). In addition, we found more patients with post-diagnostic LOG growth (50.9%) among those with low pre-diagnostic slope, compared with medium slope (35.7%) and high slope (38.2%) (Fig. 4). However, the post-diagnostic growth kinetics did not differ between the patient group with high and medium pre-diagnostic slopes.

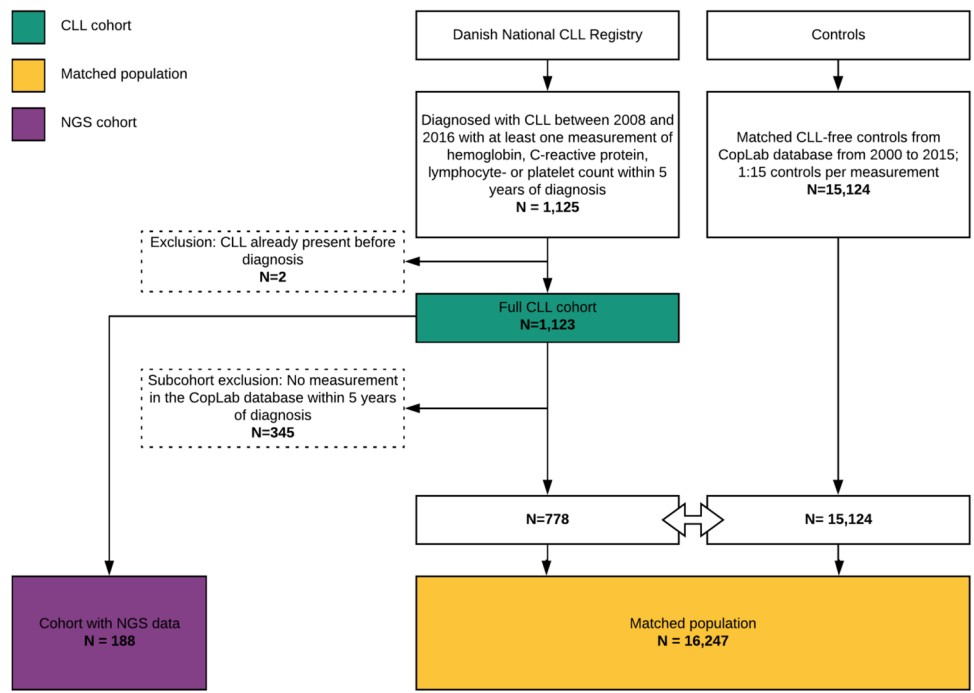

**Fig. 1 Flow chart of patients included in the study.** One thousand one hundred and twenty-five persons were diagnosed with CLL. Two persons had a wrong diagnosis date in the register. Three hundred and forty-five patients did not have any measurements in the CopLab database within 5 years of diagnosis resulting in 778 persons with CLL included in the study. The persons were matched on each biomarker with CLL-free control resulting in 15,124 controls. One hundred and eighty-eight persons had next-generation sequencing data available at the time of diagnosis. CLL chronic lymphocytic leukemia, NGS next-generation sequencing.

**Table 1 Patient characteristics.**

|  | **CLL cohort** | **Matched population** |  |
|---|---|---|---|
| Level | CLL patients | Case: CLL patients | Control: subjects without CLL |
| Number of individuals | 1123 | 778 | 15,124 |
| Age at diagnosis in years (median (IQR)) | 69 (62-77) | 69 (62-77) | 69 (62-78) |
| Sex (number of males (percentage)) | 668 (59%) | 444 (57%) | 8726 (58%) |
| Before diagnosis |  |  |  |
| Follow-up in years (median (IQR)) | 2.9 (0.1-8.1) | 7.8 (4.5-10.8) | 7.9 (4.6-10.8) |
| Absolute lymphocyte count (number) | 4073 | 3185 | 48,713 |
| Hemoglobin concentration (number) | — | 4700 | 75,432 |
| Platelet count (number) | — | 4687 | 75,348 |
| CRP (number) | — | 4263 | 76,787 |
| After diagnosis |  |  |  |
| Biomarker follow-up in years (median (IQR)) | 2.3 (0.9-4.5) | — | — |
| Absolute lymphocyte count (number) | 16,958 | — | — |
| Event time follow-up in years (median (IQR)) | 2.6 (1.2-4.7) | — | — |

*CLL* chronic lymphocytic leukemia, *IQR* interquartile range.
Patient characteristics of the persons who were followed in the CLL cohort and the matched cohort.

**Growth patterns, IGHV status, and recurrent mutations**. IGHV status and next-generation sequencing of 25 CLL driver genes were available at the time of diagnosis for 188 (16.7%) patients including 73, 103, and 10 patients LOG, IND, and EXP growth rates, respectively. Patients with LOG growth were more likely IGHV mutated (M-CLL; 64/73 patients) compared to patients with IND growth (61/103) and EXP growth (3/10), (Fig. 5). Next-generation sequencing of 25 CLL driver genes was available at the time of diagnosis in 188 (16.7%) consecutive patients including 73, 103, and 10 with LOG, IND, and EXP growth rates, respectively; two were missing[18] (Supplementary Figs. 5–8). The number of driver mutations was lowest for patients with LOG growth compared with those with IND or EXP growth (median 0 [IQR 0–1] vs median 1 [IQR 0–2] vs median 1 [IQR 0.2–1.8]; $p = 0.0014$) with LOG growth having the highest proportions of patients without driver mutations (64.4% of LOG, 39.8% of IND and 30% of EXP). Among 10 driver genes currently undergoing multicenter investigation by ERIC (ericll.org), *NOTCH1* mutations were enriched in EXP and IND growth compared with LOG growth (20.0 vs 17.5 vs 4.1%, respectively; $p = 0.047$), while *XPO1* mutations were enriched only in EXP compared with IND and LOG growth (20.0 vs 2.9 vs 2.7%; $p = 0.049$) (Fig. 5). In contrast, FISH alterations were similarly distributed in the three different growth rate groups.

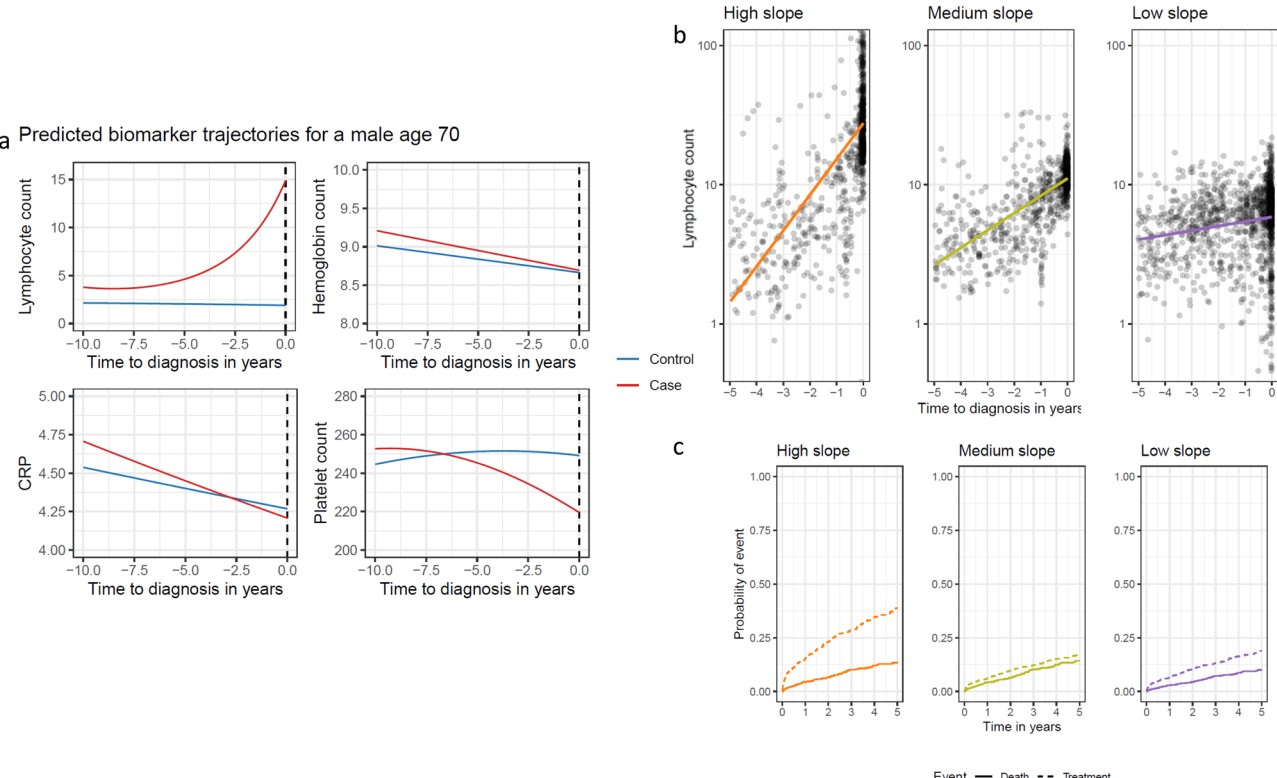

**Fig. 2 Model predictions for a 70-year-old person. a** Predictions for a 70-year-old male based on the linear regression model for absolute lymphocyte count (ALC), hemoglobin count, CRP, and platelet count. **b** Observed ALC for the three slope groups. The colored line is a linear smoother. **c** The graphs show the cumulative incidence of time to first treatment or death from time of diagnosis for the three pre-diagnostic slope groups for a male with CLL-IPI 1. CRP C-reactive protein, CLL-IPI international prognostic index for chronic lymphocytic leukemia.

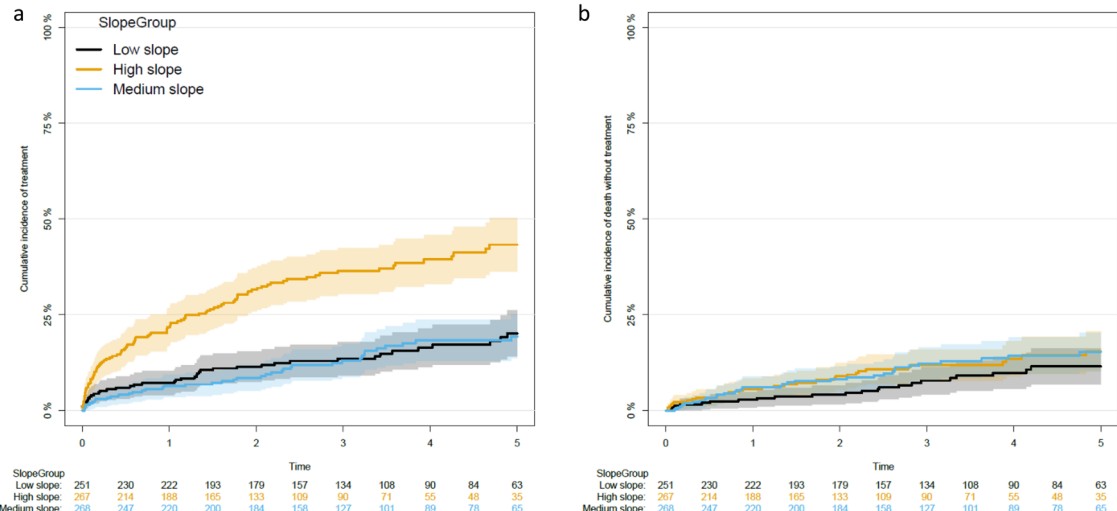

**Fig. 3 Cumulative incidence for treatment and death for the three slope groups.** Aalen–Johansen cumulative incidence estimates for the three slope groups for treatment (**a**) and death (**b**). Each patient could only have one event, that being whichever came first. Time zero is the time of diagnosis for all patients. The shaded areas for both figures are 95% confidence intervals.

## Discussion

We know that CLL is detectable years before the diagnosis of CLL by flow cytometry and DNA-based methods[22]. We here demonstrate that the kinetics of the CLL clone in terms of lymphocytosis prior to diagnosis of CLL carries important prognostic information and correlates with underlying genetic aberrations by which the disease evolves. We identified that the average accumulation of CLL cells is detectable and measurable 3 years before diagnosis and the slope of the lymphocyte count prior to diagnosis is independently associated with the risk of treatment. In addition, different growth patterns prior to diagnosis correlate to some degree with the growth kinetics after diagnosis as described by Gruber and colleagues[12]. Furthermore, both pre-diagnostic and post-diagnostic growth kinetics could be correlated with genetic phenotype at diagnosis of CLL.

Over the past three decades, the results of several studies have shown that MBL and CLL are closely related consecutive

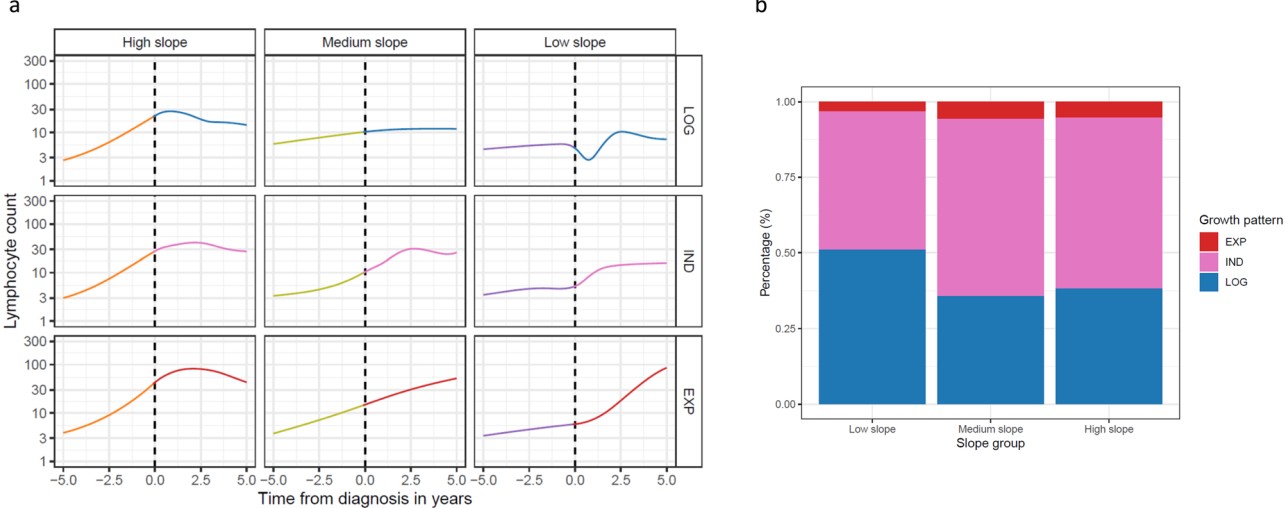

**Fig. 4 Patterns and associations for the growth and slope groups. a** Smoothed trajectory of the development of lymphocyte count before and after diagnosis in the three slope groups and the three growth patterns. **b** Distribution of the growth patterns between the lymphocyte slope groups. LOG logistic, IND indeterminate, EXP exponential-like.

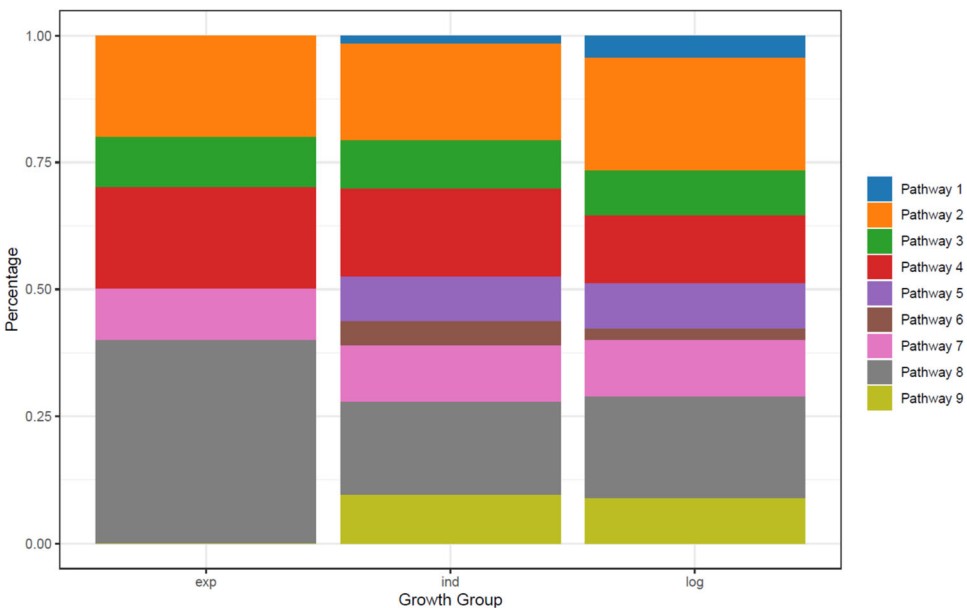

**Fig. 5 Percentage of persons with driver mutations altering a signaling pathway.** Pathway 1; Chromatin modification: ZMYM3, Pathway 2; DNA damage: ATM, POT1, TP53, CCND2, Pathway 3; Apoptosis: BIRC3, TP53, Pathway 4; Notch signaling: FBXW7, NOTCH1, Pathway 5; Inflammatory pathway: BIRC3, DDX3X, MYD88, TRAF3, Pathway 6; MAPK, ERK, BRAF, KRAS, NRAS, TRAF3, Pathway 7; NFkB pathway: BCOR, BIRC3, EGR2, IRF4, NFKBIE, Pathway 8; RNA and ribosomal processing: DDX3X, MED12, NXF1, RPS15, SF3B1, XPO1, ZNF292, Pathway 9; MYC related: FBXW7, MGA.

conditions with MBL being present years prior to diagnosis of CLL. Specifically, MBL has been demonstrated up to 77 months in advance of CLL diagnosis and serum immunoglobulin abnormalities up to 112 months (or 9 years) before CLL diagnosis[22, 23]. However, MBL does not always progress to CLL, as evidenced by the high prevalence of MBL among the elderly and among relatives of CLL patients in particular. MBL can be demonstrated in approximately 5% at age 40+ years and in 12% of persons above 65 years of age[3, 24, 25]. While it is currently not possible to predict whether a person with MBL eventually progresses to CLL, the detailed temporal trajectory of pre-diagnostic lymphocytosis is consistent with the notion that the clone develops many years before diagnosis and the kinetics of the pre-diagnostic lymphocytosis hold prognostic information[22, 26]. This is in line with our recent report that overall antimicrobial use

gradually increases among CLL patients from six years before diagnosis, thus emphasizing the co-occurrence of the pre-malignant clone and immune dysfunction[27].

Our approach to integrating pre-diagnostic biochemistry data for prognostication is new, and it may prove helpful on top of the increasing amount of data available for each patient to personalize management from the time of diagnosis of CLL. We show that the patient-specific pre-diagnostic lymphocyte slope is significantly associated with risk of treatment independent of CLL-IPI risk factors and gender. The addition of patient-specific slopes improved the prognostic model for both treatment and death. A recently published model showed that IGHV mutational status, ALC of >15,000 per µL, and nodal involvement were a robust prognostic model for early-stage asymptomatic patients with CLL[9]. This finding is in line with our addition of slope groups,

and emphasizes the importance of ALC and the doubling time even in early-stage patients.

The concept of lymphocyte doubling time and lymphocyte kinetics is intuitively important and not new - numerous studies have reported doubling time to be prognostically important after diagnosis of CLL and is even an indication for treatment[2]. Interestingly, we found slope groups to be prognostic for treatment, but not survival, which highlights that the number of CLL cells is important for the treatment but not treatment response, where cytogenetic of the CLL cells are more important, i.e., del17p and IGHV mutational status etc[2].

Although CLL cells are present and diagnosis is based on their presence in the peripheral blood, the growth of CLL cells mainly occurs in the lymph nodes[10]. Despite the complex interplay between the three compartments of bone marrow, lymph nodes, and the blood, our results are a simplification of disease dynamics that independently correlate with time to first treatment. Thus, for clinicians with access to pre-diagnostic ALC, the here developed model can be accessed here and help better personalize the management of asymptomatic patients with CLL in the everyday clinic.

The app is trained exclusively on data retrieved before diagnosis. However, since the time point of diagnosis of CLL is somewhat arbitrary and the spectrum of where in the disease stage a patient is present is very wide, we believe that the app can be extrapolated to treatment-naive patients also after diagnosis.

To assess the impact of pre-diagnostic lymphocyte kinetics on post-diagnostic growth patterns, we compared our slope groups to the post-diagnostic growth patterns as reported by Gruber and colleagues[12]. Although not near perfect correlation, we did find that patients with a pre-diagnostic high or medium slope had a higher risk of post-diagnostic EXP growth while patients with a low slope had a higher chance of subsequent LOG growth. Additionally, we confirm that the cohort with the EXP growth pattern included more patients with unmutated IGHV, and the LOG growth group included more IGHV mutated patients. We also observed a higher number of activated pathways and mutations among patients with EXP and IND growth as compared to patients with LOG growth. The disease is dynamic, and the slope and growth may indeed change over the course of the disease as the CLL clone can accumulate more somatic mutations[28]. Even so, it was reassuring that we could verify results published by Gruber and colleagues[12].

In this study, driver mutations were enriched in patients with EXP growth confirming a more aggressive and progressive CLL clone for patients with EXP compared with LOG growth. While other studies have reported a shorter time to first treatment in patients carrying more driver mutations, our study demonstrates that the number of driver mutations correlates with faster lymphocyte growth resulting in shorter lymphocyte doubling time[18, 19]. This may in part explain the mechanistic link between tumor load, as here simply modeled by lymphocyte kinetics, and time to first treatment. Specifically, NOTCH1 and XPO1 mutations were enriched in patients with EXP growth. However, this may be a chance finding as no single gene was correlated with EXP growth following correction for multiple testing, underscoring the importance of the cumulative tumor growth rate as assessed by lymphocyte kinetics here. Even so, two of ten patients with EXP growth had neither U-CLL, high-risk FISH (i.e., +12, 11q− and 17p−) nor detectable driver mutations indicating that non-genetic factors including microenvironmental interactions and genetic aberrations not covered by the here applied analyses likely impact lymphocyte growth as well[29].

One of the major strengths of our study includes its unique population-based retrospective design with pre-diagnostic blood samples. However, the information in the CopLab database was not collected for research purposes, and persons were not systematically tested. In addition, we have no knowledge of the indication for which the general practitioners ordered the blood tests. The clinical assumption is, however, that CRP and complete blood counts are measured based on suspicion of infection, due to other chronic diseases like hypertension and diabetes with regular blood workup or as a routine check. This means that our results cannot easily be implemented prospectively as our results are based on clinical decisions made by the general practitioners. However, clinicians with similar *biased* data available will be able to use our model for prognostication and personalized management of newly diagnosed patients with CLL. As mentioned, we excluded observations from some analyses due to missing information in the biomarker variables. Hence, the validity of those results hinges on the assumption that the excluded observations would not have altered them had they had complete information. Lastly, small lymphocytic lymphoma (SLL) may also bias our results. In Denmark, 5–10% of all CLL present as SLL, for these patients the ALC would by definition be below 5000 clonal B cells per μL at diagnosis, which would bias our estimates of slope groups toward the null.

The clinical challenge in the coming years is to educate low-risk patients and better prognosticating newly diagnosed patients. In Denmark, 30–40% will never require treatment, nevertheless, Levin and colleagues found depression, anxiety, and quality of life (QoL) were similar in "watch and wait" vs actively treated CLL[30]. Patients diagnosed for more than 6 years had a worse physical QoL, but their social and emotional QoL were like those of newly diagnosed patients[30]. Thus, further research is needed before we may introduce the identification of MBL patients at high risk for early interventions to be tested in clinical trials while ending watch and wait for some patients already diagnosed with CLL, as we have currently implemented[31]. We should aim to gain an understanding of biological features that can identify those (few) cases that are at risk of developing a progressive CLL with treatment need or significant immune dysfunction warranting further interventions. Clinical MBL and CLL seem to be a continuum and both conditions are a mixture of patients whose life expectancy will be affected by the disease; but also, individuals who will never develop any clinical signs or symptoms of CLL. In general, about one-third of patients with CLL will never require treatment, and many may have survival identical to age-matched unaffected individuals. The remaining patients will eventually need treatment at variable times after presentation, reflecting different degrees of disease aggressiveness. The ideal main distinction should thus be between potentially progressive and non-progressive cases rather than based upon a numerical threshold. A recent prognostic model for asymptomatic CLL patients found that low-risk CLL patients had 5-year cumulative risk of treatment of mere 8.4%[9]. The model included only three variables, (1) nodal involvement, (2) IGHV mutational status, and (3) CLL cell count ($<15 \times 10^9$/L).

Going forward, the approach of including all available (para) clinical data for the development of predictive models, as employed in CLL-TIM, would eventually allow us to provide decision support based on the prediction of different outcomes during the disease course for individual patients[32]. Consequently, we may be on the right track to identify whom to diagnose with leukemia and whom to diagnose with age-related MBL.

## Data availability

Restrictions apply to the availability of the underlying data sets, which were used under specific conditions for the current study. The reason for not making the data set publicly available is that Danish law restrict publication of any individual-level data and group with less than five persons. Moreover, according to Danish legislation, sequencing data cannot be fully anonymized, and data cannot be shared in a repository. All Danish registry data can be obtained through application to the relevant data agencies.

Sequencing data used were previously published[18]. Source data for the main figures are available at https://github.com/masger/cllprognosis.

## Code availability

Analytical methods and source code are stored https://github.com/masger/cllprognosis. The app is available at https://clllab.shinyapps.io/cllprognosis/. The source code (https://doi.org/10.5281/zenodo.6470207) for the app is available at: https://github.com/masger/cllprognosis[33].

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

## Acknowledgements

This work is supported in part by the Research Foundation of Rigshospitalet and Danish National Research Foundation [grant 126]. C.U.N. is supported by Novo Nordisk Foundation grant [NNF16OC0019302] and the Danish Cancer Society.

## Author contributions

M.A.A., M.K.G., C.U.N., and C.B. analyzed, and interpreted data. M.A.A. drafted the manuscript. M.K.G. and V.S. performed the statistical analyses. C.L.A. co-designed the CopLab database. All authors critically revised the manuscript for important intellectual content and approved the final version to be submitted.

## Competing interests

C.U.N. received grants/consultancy fees from AbbVie, Janssen, Gilead, AstraZeneca, CSL Behring, Takeda, Octapharma, Roche, and Novartis outside of this study. M.A.A., M.K.G., C.B., V.S., and C.L.A. report no conflict of interest.
