## [Peer Review File · Communications Medicine]

Reviewers' comments:

Reviewer #1 (Remarks to the Author):

1. Brief summary of the manuscript

In the presented manuscript, Andersen et al. explore the impact of pre-diagnostic dynamics in lymphocytosis on the risk of need for CLL treatment and the risk of death from CLL. To this end, they investigated 10-year trajectories of hemoglobin, C-reactive protein (CRP), ALC and platelet count prior to diagnosis of CLL based on a Danish matched case-control population, with 15 random controls (CLL-free individuals) per case (CLL patients). In total, the matched case-control group contains 778 CLL patients and 15,124 CLL-free controls.

Based on their data, they also provide an online app, where users can enter the CLL IPI, sex, date of diagnosis, as well as two or more values of pre-diagnostic lymphocyte counts. The app then estimates probability of CLL treatment and death from CLL.

Finally, the authors assess post-diagnostic CLL growth patterns for the 548 patients in their cohort with at least 3 assessments using a previously described model (Gruber et al., Nature 2019), and link these patterns with IGHV status and recurrent mutations (also grouped by pathways) for 188 patients.

2. Overall impression of the work

Andersen et al. provide a straightforward, well-designed and presented work based on data from an impressively big case-control cohort. They corroborate findings from a previous work on this cohort and apply and extend a previously published model for CLL growth dynamics, by this means presenting valuable new findings and applications. The included web-application is easy to navigate and might be useful and expandable for clinical assessment and/or subsequent scientific work.

3. Specific comments, with recommendations for addressing each comment

A

The model specifies pre-diagnostic ALC, however the app also appears to work when values after diagnosis date are entered. Pre-diagnostic values are often not available, however mostly there are several observations after diagnosis. Could the models' applicability be extended to include these?

B

It would be great if the graphs in the app could include information about statistical confidence (e.g. shades for the confidence intervals behind the lines)

C

A table or graph illustrating the mixed model for prognostic significance, with the other parameters included, would be helpful.

Reviewer #2 (Remarks to the Author):

Andersen and colleagues report on a large cohort of patients with chronic lymphocytic leukemia

(CLL) who have pre-diagnostic information available from Denmark. The authors demonstrate the value of change in absolute lymphocyte count (ALC) as an added variable at the time of diagnosis that can help predict time to first treatment (TFT). I have several comments for the authors:

1. Title: the word “the” in “....predicts with the time to treatment” appears out of place and may be removed.
2. An important limitation of the work is that ALC does not take into account a small lymphocytic lymphoma (SLL) type of clinical presentation – and this should be acknowledged in the discussion section.
3. Can the authors elaborate on why they chose C-reactive protein (CRP) as an “important biomarker of interest” – Line 73? Are there data to support its use as a biomarker in CLL – either with respect to time to first treatment or overall survival (OS)?
4. The results presented in Lines 94-100 are a little confusing. Why are data for a representative patient with CLL with age of 70 years presented? Can the authors instead present cumulative data of all CLL patients included in this study at the time of diagnosis? In addition, criteria for diagnosis of CLL requires at least 5000 clonal B lymphocytes per μL of blood. The authors indicate the ALC was 15,000 per μL . Can the authors confirm that this corroborated to a clonal B cell count of greater than 5000 cells per μL in all patients eventually diagnosed with CLL? In addition, please add the CLL-IPI risk score for all patients.
5. The authors did not include any information about the changes in the biomarkers of interest in the control population – can this information be added. Although it is likely that the control population had no changes in these biomarkers; it would be nice to have this mentioned and referenced somewhere in the manuscript.
6. The authors mention that the CLL-IPI is a poor predictor of TFT in newly diagnosed CLL patients. They describe a new and more recent prognostic model that includes only three variables (CLL; IPS-E): are the authors able to adjust the results of TFT using low, medium and high slope of ALC growth prior to CLL diagnosis to predict TFT.
7. In addition, please provide a Table that shows the HR for all variables included in the multivariable model, number of observations used, and the corresponding 95% CI and p-values – including the non-significant p-value for the difference between low and medium slope groups to predict TFT. Additionally, it is unclear if each of these slopes as univariable parameters predict TFT in all CLL patients?
8. Figure 3 shows that the ALC in patients with pre-diagnosis high slope and EXP growth following diagnosis of CLL – here the ALC appears to be decreasing after a diagnosis of CLL. Conversely, in patients with pre-diagnosis low slope, the ALC appears to increase quickly after a diagnosis of CLL in EXP growth after CLL diagnosis. Both these scenarios are unclear - can the authors clarify how many patients are in each of these 9 boxes?
9. The authors describe the importance of pre-diagnostic ALC and its ability to add incremental value to CLL-IPI and sex in predicting TFT. However, the p-value for the difference in low and medium slope was not significant. What is the c-statistic for the CLL-IPI alone in predicting TFT and how much does the pre-diagnosis slope add to improving this c-statistic?

Reviewer #3 (Remarks to the Author):

In this manuscript, the relationship between patient lymphocytosis trajectory and CLL is studied.

1. since the lymphocytosis is longitudinally measured, to characterize the relationship between

patient lymphocytosis trajectory and CLL the joint models of longitudinal data analysis and the Cox proportional hazards model should be employed.

2. since the slopes of lymphocytosis changes before CLL are used as a biomarker for CLL diagnosis, the corresponding sensitivity and specificity (or AUC) should be estimated.

Then, the statistical analysis should be improved.

Referee expertise:

Referee #1: CLL clinical, lymphocyte dynamics

Referee #2: CLL clinical, biomarkers

Referee #3: Stats, prognostics

Reviewers' comments:

Reviewer #1 (Remarks to the Author):

1. Brief summary of the manuscript

In the presented manuscript, Andersen et al. explore the impact of pre-diagnostic dynamics in lymphocytosis on the risk of need for CLL treatment and the risk of death from CLL. To this end, they investigated 10-year trajectories of hemoglobin, C-reactive protein (CRP), ALC and platelet count prior to diagnosis of CLL based on a Danish matched case-control population, with 15 random controls (CLL-free individuals) per case (CLL patients). In total, the matched case-control group contains 778 CLL patients and 15,124 CLL-free controls.

Based on their data, they also provide an online app, where users can enter the CLL IPI, sex, date of diagnosis, as well as two or more values of pre-diagnostic lymphocyte counts. The app then estimates probability of CLL treatment and death from CLL.

Finally, the authors assess post-diagnostic CLL growth patterns for the 548 patients in their cohort with at least 3 assessments using a previously described model (Gruber et al., Nature 2019), and link these patterns with IGHV status and recurrent mutations (also grouped by pathways) for 188 patients.

2. Overall impression of the work

Andersen et al. provide a straightforward, well-designed and presented work based on data from an impressively big case-control cohort. They corroborate findings from a previous work on this cohort and apply and extend a previously published model for CLL growth dynamics, by this means presenting valuable new findings and applications. The included web-application is easy to navigate and might be useful and expandable for clinical assessment and/or subsequent scientific work.

3. Specific comments, with recommendations for addressing each comment

A

The model specifies pre-diagnostic ALC, however the app also appears to work when values after diagnosis date are entered. Pre-diagnostic values are often not available, however mostly there are several observations after diagnosis. Could the models' applicability be extended to include these?

We have updated the app to only allow values before diagnosis. We agree with the reviewer that the model's applicability could be extended by also modelling on data after diagnosis. However, we are limited by the fact that patients in this study were not exclusively low risk patients and some would therefore need treatment at time of diagnosis or close thereafter; thus changing the lymphocyte dynamics and formally not allowing to model based on data after diagnosis. However, since the diagnosis of CLL is very heterogenous and the time point of diagnosis is somewhat random, one could extrapolate a treatment-

naïve patient to be “at diagnosis” also after diagnosis. We have added a sentence in the manuscript to further this point and updated the information in the app as well.

The app is trained exclusively on data retrieved before diagnosis. However, since the time point of diagnosis of CLL is somewhat arbitrary and the spectrum of where in the disease stage a patient present with is very wide, we believe that the app can be extrapolated to treatment naïve patients also after diagnosis.

B

It would be great if the graphs in the app could include information about statistical confidence (e.g. shades for the confidence intervals behind the lines)

We agree, and we have added CIs in the app.

C

A table or graph illustrating the mixed model for prognostic significance, with the other parameters included, would be helpful.

Thank you for this proposal, we have added a table in the supplementary (Table S10) showing CI for all parameters included in the model. This shows the contribution of each parameter in the linear mixed model and the competing risk Cox proportional hazards models.

Reviewer #2 (Remarks to the Author):

Andersen and colleagues report on a large cohort of patients with chronic lymphocytic leukemia (CLL) who have pre-diagnostic information available from Denmark. The authors demonstrate the value of change in absolute lymphocyte count (ALC) as an added variable at the time of diagnosis that can help predict time to first treatment (TFT). I have several comments for the authors:

1. Title: the word “the” in “...predicts with the time to treatment” appears out of place and may be removed.

Thank you. Corrected.

2. An important limitation of the work is that ALC does not take into account a small lymphocytic lymphoma (SLL) type of clinical presentation – and this should be acknowledged in the discussion section.

Yes, that is correct. The Danish CLL register includes SLL as well (in line with the WHO definition). This means that persons with SLL are also in this cohort. Controls were cancer free at the time of the index date, nevertheless, persons with SLL would contribute very little to the mean lymphocyte growth and would probably bias our estimates towards null. We have added a sentence about this in the limitations in the discussion section.

Lastly, small lymphocytic lymphoma (SLL) may also bias our results. In Denmark, 5-10 % of all CLL present as SLL, for these patients the ALC would by definition be below 5000 clonal B-cells per μ L at diagnosis, which would bias our estimates of slope groups towards null.

3. Can the authors elaborate on why they chose C-reactive protein (CRP) as an “important biomarker of interest” – Line 73? Are there data to support its use as a biomarker in CLL – either with respect to time to first treatment or overall survival (OS)?

We agree with the reviewer, that CRP is not specific/correlated to CLL in contrast to ALC and TRC. However, CRP may correlate with lymphocyte/leukocyte changes due to infection and has been predictive in many cancers. For example, in a study from 2009, where the authors found CRP to be associated with many types of cancer.

4. The results presented in Lines 94-100 are a little confusing. Why are data for a representative patient with CLL with age of 70 years presented? Can the authors instead present cumulative data of all CLL patients included in this study at the time of diagnosis? In addition, criteria for diagnosis of CLL requires at least 5000 clonal B lymphocytes per μL of blood. The authors indicate the ALC was 15,000 per μL . Can the authors confirm that this corroborated to a clonal B cell count of greater than 5000 cells per μL in all patients eventually diagnosed with CLL? In addition, please add the CLL-IPI risk score for all patients.

We present data on one modelled patient to show how the parametric model fits the data for an individual example patient. But we agree with the reviewer that the non-parametric plots are equally important. We have thus included a plot for cumulative incidence for all patients in the supplementary (Figure S7).

As mentioned earlier in response to reviewer 1, the time point of diagnosis of CLL is somehow arbitrary. We have included a histogram of the ALC at diagnosis in the supplementary (Figure S8). From the histogram, we can also deduct that 5-10 % of our cases were diagnosed with SLL rather than CLL, which is of course an important bias to our study as detailed in response to issue 2 above.

Thank you for pointing out the need for providing the CLL-IPI scores, we have added a table with those for all patients (Table S9).

5. The authors did not include any information about the changes in the biomarkers of interest in the control population – can this information be added. Although it is likely that the control population had no changes in these biomarkers; it would be nice to have this mentioned and referenced somewhere in the manuscript.

Agree. We have added the following sentence:

At time of diagnosis (year 0), patients with CLL had a mean estimated ALC of 15,000 per μL compared with 2,400 per μL for controls in accordance with the criteria for CLL diagnosis requiring at least 5,000 clonal lymphocytes per μL . The gradual increase in ALC for CLL patients could be detected three to seven years before the diagnosis of CLL, whereas the ALC for controls were constant. For CRP and hemoglobin concentration, no differences were observed between cases and controls up until one year before the diagnosis of CLL, with both hemoglobin and CRP declined with increasing age. However, mean platelet count was inversely correlated with mean ALC. Figure 2.

6. The authors mention that the CLL-IPI is a poor predictor of TFT in newly diagnosed CLL patients. They describe a new and more recent prognostic model that includes only three variables (CLL; IPS-E): are the authors able to adjust the results of TFT using low, medium and high slope of ALC growth prior to CLL diagnosis to predict TFT.

The model IPS-E includes ALC (above 15 000 cells per μL), nodal involvement and IGHV mutational status. Our model includes the slope for the estimate of prognosis while the the CLL-IPI also takes IGHV mutational

status and to some degree nodal involvement into account (clinical stage). Therefore, we believe that ALC growth prior to CLL would also improve IPS-E, but we cannot formally test this, as we do not have information on nodal involvement for the cohort. But we agree with the reviewer and have added a sentence to the discussion.

A recent published model showed that IGHV mutational status, ALC of >15,000 per μ L and nodal involvement was a robust prognostic model for early stage asymptomatic patients with CLL [9]. This finding is in line with our addition of slope groups, and emphasizes the of importance of ALC and the doubling time even in early stage patients.

7. In addition, please provide a Table that shows the HR for all variables included in the multivariable model, number of observations used, and the corresponding 95% CI and p-values – including the non-significant p-value for the difference between low and medium slope groups to predict TFT. Additionally, it is unclear if each of these slopes as univariable parameters predict TFT in all CLL patients?

We now included this in the supplementary (Table S10).

8. Figure 3 shows that the ALC in patients with pre-diagnosis high slope and EXP growth following diagnosis of CLL – here the ALC appears to be decreasing after a diagnosis of CLL. Conversely, in patients with pre-diagnosis low slope, the ALC appears to increase quickly after a diagnosis of CLL in EXP growth after CLL diagnosis. Both these scenarios are unclear - can the authors clarify how many patients are in each of these 9 boxes?

The reviewer points towards some weird fitting due to low numbers in the two models in Figure 3. The weird results for the two models in Figure 3 are probably because of the low number of patients in these strata. In the group High Slope and EXP growth we have 8 patients. In low slope and EXP growth we had 5 patients. The number of subject in each growth and slope group can be found in the table below.

Number of subjects (nid) by their growth and slope group.

Slopegroup	group	nid	pct
High slope	exp	8	2.83
High slope	ind	86	30.39
High slope	log	58	20.49
High slope	mis	5	1.77
High slope	NA	126	44.52
Low slope	exp	5	1.77
Low slope	ind	70	24.73
Low slope	log	78	27.56
Low slope	NA	130	45.94
Medium slope	exp	8	2.84
Medium slope	ind	82	29.08
Medium slope	log	50	17.73
Medium slope	NA	142	50.35
NA	exp	4	1.45
NA	ind	39	14.18
NA	log	60	21.82
NA	mis	1	0.36
NA	NA	171	62.18

9. The authors describe the importance of pre-diagnostic ALC and its ability to add incremental value to CLL-IPI and sex in predicting TFT. However, the p-value for the difference in low and medium slope was not significant. What is the c-statistic for the CLL-IPI alone in predicting TFT and how much does the pre-diagnosis slope add to improving this c-statistic?

Thanks for pointing out the need to clarify this. We have compared CLL-IPI vs CLL-IPI with slope group. Using a likelihood ratio test to compare the two models, the model including slope group were significantly better. As you mention, a more direct way to compare the predictions would be to look at the difference in AUC (c-statistics, delta AUC in the figures below) over time between the Cox Models where the slope group was not included (Cox models without slope group) and the Cox Models where the slope group was included (Cox models). A negative difference in AUC, thereby, means a smaller AUC at a given timepoint for the Cox models without slope group compared to the Cox models with slope group. For treatment the addition of slope groups improved the AUC at year 1 with 2.2 percentage points, and 3.8 percentage points at year 2.

Treatment

For death the AUC was improved with 4.5 for the first year and 4.1 second year.

We have added the following paragraph to the manuscript:

The addition of slope group improved the model predictions (compared to CLL-IPI alone) for both treatment and death throughout the period, e.g. at 1 year after diagnosis the AUC was improved with 2.2 percentage points (95% CI [-1.5, 6.0]) for treatment and with 4.5 percentage points (95% CI [0.2, 8.9]) for death. However, as expected, the predictive effect of the pre-diagnostic ALC slope group declined over time.

Reviewer #3 (Remarks to the Author):

In this manuscript, the relationship between patient lymphocytosis trajectory and CLL is studied.

1. since the lymphocytosis is longitudinally measured, to characterize the relationship between patient lymphocytosis trajectory and CLL the joint models of longitudinal data analysis and the Cox proportional hazards model should be employed.

Our objective was to use prediagnostic lymphocyte measurements to predict risk of needing treatment from time of diagnosis (time 0). Hence, our objective gives us a setup that is not the same as the setup where joint models are usually employed, e.g. as you mention when biomarkers are longitudinally measured after time 0. Thus we hope you agree that the here used methods are appropriately used.

2. since the slopes of lymphocytosis changes before CLL are used as a biomarker for CLL diagnosis, the corresponding sensitivity and specificity (or AUC) should be estimated.

As mentioned above, we have used the ALC growth pattern as a predictor for CLL prognosis and by adding this information we were able to improve the prognostic index with 4.5 percentage points at year 1 for

death and 2.2 percentage points at year 1 for treatment. Thus, we are not using the slopes as a biomarker for CLL diagnosis but for prognosis after diagnosis of CLL.

Then, the statistical analysis should be improved.

During the revision we have added text, tables and figures to the manuscript and the appendix which we believe have improved and clarified the methods considerably. We are thankful for the help and comments from reviewers during this process and believe that the manuscript and its analyses have hereby been significantly improved.

Reviewers' comments:

Reviewer #1 (Remarks to the Author):

My points have been addressed sufficiently. I have no further comments.

Reviewer #2 (Remarks to the Author):

Thank you for addressing my comments. I appreciate it.

I would like to request the authors consider some additional minor requests:

1. Supplementary Figure 7 should be moved to the main manuscript. By showing one representative patient, you risk overfitting the data to show what is not the reality for the entire cohort.
2. In Supplementary Figure 7, it is quite interesting that the time to first therapy curves for high and medium slope track each other; but are quite divergent for overall survival. The authors should consider discussing this further.
3. Thank you for including the CLL-IPI distribution in Table S9. The % are listed as 0.59%, etc.; it should be 59%, etc. instead.
4. Given the time-to-event nature of the analysis – including both TTFT and OS; I believe using likelihood ratio to demonstrate the added benefit of the pre-diagnosis ALC slope to CLL-IPI is incorrect. Using Cox models and AUC description as provided in the response letter should be instead included in the main manuscript. Along the same lines, the Figures included in the response to reviewer document which show the %improvement in AUC over time should be added as supplementary figures and corresponding text and discussion should be included in a revised manuscript.
5. The authors assert in the discussion that we should not “implement screening protocols for the general population to create an endless cohort of potential patients with MBL”. I think this is a very strong statement and should be tempered down – since there are data that show individuals with MBL have a higher risk of death compared to those without MBL (PMID: 29567775) and increased risk of serious infections (PMID: 32203143). I would consider these efforts complementary – where the combination of screening for a high-risk condition and additional work as elegantly done by the authors would best identify those individuals with highest risk of adverse consequences with small B cell clones.

Reviewer #3 (Remarks to the Author):

In this manuscript, a pre-diagnosis for predicting the time to treatment and death in patients with chronic lymphocytic leukemia is developed based on trajectories of lymphocytosis. The patients are divided into three groups with ALC growth rates: low slope, medium slope and high slope. I have the following questions:

1. The differences of Patients' demographic and clinical variables (such as gender, age, etc.) among the three groups should be given.
2. The estimated survival curves of the time to treatment and death in patients in each group should be given, and their comparison analysis is also very important.

3. Please give the correlation analysis of trajectories of ALC, haemoglobin count, CRP and Platelet count.

4. Since the patients' trajectories of ALC, haemoglobin count, CRP and Platelet count are longitudinally observed, to develop a pre-diagnostic model for the time to treatment and death in patients, the joint modelling of survival analysis and longitudinal data analysis can be employed.

Reviewers' comments:

Reviewer #1 (Remarks to the Author):

My points have been addressed sufficiently. I have no further comments.

Reviewer #2 (Remarks to the Author):

Thank you for addressing my comments. I appreciate it.

I would like to request the authors consider some additional minor requests:

1. Supplementary Figure 7 should be moved to the main manuscript. By showing one representative patient, you risk overfitting the data to show what is not the reality for the entire cohort.

We have moved S7 to the main manuscript, showing cumulative incidences for all patients, including a risk table, so readers can examine the number at risk at a given time point. We agree with the reviewer that it makes sense to include this figure as well.

2. In Supplementary Figure 7, it is quite interesting that the time to first therapy curves for high and medium slope track each other; but are quite divergent for overall survival. The authors should consider discussing this further.

The reviewer points something out which contrasts our finding. I have talked with the statistician and she have accidentally labelled the figures wrong. The high slope group have shorter time to first treatment but comparable overall survival. We have updated the figure with correct labels. Intuitively, this observation makes sense since a fast doubling time is a treatment criterion. In addition, more CLL cells means an increased disease burden. However, since the overall survival does not follow the same pattern, the slope group is a poor predictor for overall survival. This may indicate that it is not the amount of CLL cells or the doubling time that are important for the treatment outcome, rather the molecular markers of the specific CLL cells, i.e. del17p and IGHV mutational status etc.

We have included the following paragraph in the discussion.

Interestingly, we found slope groups to be prognostic for treatment, but not survival, which highlight that the number of CLL cells are important for the treatment but not treatment response, where cytogenetic of the CLL cells are more important, i.e. del17p and IGHV mutational status etc. [2].

3. Thank you for including the CLL-IPI distribution in Table S9. The % are listed as 0.59%, etc.; it should be 59%, etc. instead.

Thank you, we have corrected this.

4. Given the time-to-event nature of the analysis – including both TTFT and OS; I believe using likelihood ratio to demonstrate the added benefit of the pre-diagnosis ALC slope to CLL-IPI is incorrect. Using Cox models and AUC description as provided in the response letter should be instead included in the main manuscript. Along the same lines, the Figures included in the response to reviewer document which show the %improvement in AUC over time should be added as supplementary figures and corresponding text and discussion should be included in a revised manuscript.

Agree. We have added a sentence to the results and discussion. We have added the AUC plots to supplementary, thus making it possible for readers to address how the models compare over time.

Results:

The addition of slope group improved the model predictions (compared to CLL-IPI alone) for both treatment and death throughout the period, e.g. at 1 year after diagnosis the AUC was improved with 2.2 percentage points (95% CI [-1.5, 6.0]) for death and with 4.5 percentage points (95% CI [0.2, 8.9]) for treatment. However, as expected, the predictive effect of the pre-diagnostic ALC slope group declined over time (Figure S9 and S10).

Discussion:

The addition of patient-specific slopes improved the prognostic model for both treatment and death.

5. The authors assert in the discussion that we should not “implement screening protocols for the general population to create an endless cohort of potential patients with MBL”. I think this is a very strong statement and should be tempered down – since there are data that show individuals with MBL have a higher risk of

death compared to those without MBL (PMID: 29567775) and increased risk of serious infections (PMID: 32203143). I would consider these efforts complementary – where the combination of screening for a high-risk condition and additional work as elegantly done by the authors would best identify those individuals with highest risk of adverse consequences with small B cell clones.

We agree with the reviewer that patients with MBL have increased risk of death and infections, but for now there are no treatment options or guidelines for management for these patients/persons, and thus many patients end up in routine clinical checkups. In Denmark, 30-40 % will never require treatment. Levin and colleagues found depression, anxiety, and quality of life (QoL) were similar in “watch and wait” versus actively treated CLL. Patients diagnosed for more than 6 years had a worse physical QoL, but their social and emotional QoL were like those of newly diagnosed patients. (DOI: 10.1016/j.genhosppsy.2007.01.014)

We think the discussion is very important, however we have moderated and changed the sentence based on your valuable input.

The clinical challenge in the coming years is to educating low risk patients and better prognosticating newly diagnosed patients. In Denmark, 30-40 % will never require treatment, nevertheless Levin and colleagues found depression, anxiety, and quality of life (QoL) were similar in “watch and wait” versus actively treated CLL[23]. Patients diagnosed for more than 6 years had a worse physical QoL, but their social and emotional QoL were like those of newly diagnosed patients [23]. Thus, further research is needed before we may introduce identification of MBL patients at high risk for early interventions to be tested in clinical trials while ending watch and wait for some patients already diagnosed with CLL, as we have currently implemented[24].

Reviewer #3 (Remarks to the Author):

In this manuscript, a pre-diagnosis for predicting the time to treatment and death in patients with chronic lymphocytic leukemia is developed based on trajectories of lymphocytosis. The patients are divided into three groups with ALC growth rates: low slope, medium slope and high slope. I have the following questions:

1. The differences of Patients' demographic and clinical variables (such as gender, age, etc.) among the three groups should be given.

We agree. We have included another baseline characteristic for the three slope groups in the Supplementary S9.

2. The estimated survival curves of the time to treatment and death in patients in each group should be given, and their comparison analysis is also very important.

Agree. Please see response to reviewer 2, point 1 and 2.

3. Please give the correlation analysis of trajectories of ALC, haemoglobin count, CRP and Platelet count.

We have made correlations for the ALC, haemoglobin count, CRP and platelet count and plotted it for controls and cases. Please see Figure S9. Correlation between ALC, CRP, platelet count and haemoglobin concentration prior to diagnosis. Correlations were calculated based on a randomly selected observation for each person for cases and controls. Due to the non-linear relationship between some of the biomarkers Kendall's tau was used.

4. Since the patients' trajectories of ALC, haemoglobin count, CRP and Platelet count are longitudinally observed, to develop a pre-diagnostic model for the time to treatment and death in patients, the joint modelling of survival analysis and longitudinal data analysis can be employed.

We agree that we indeed could have used our data to model the relationship between ALC and time to treatment and death from the time of CLL diagnosis and onwards using a joint-model. To estimate the joint model we would use the ALC measurements after the CLL diagnosis. The model could potentially also be adjusted for CLL measurements prior to diagnosis by using the approach we are currently using or include other biomarkers as suggested. The joint-model would enable us to make dynamic predictions of the risk of treatment and death from the time of diagnosis and during follow-up [Rizopoulos, D. (2012). Joint models for longitudinal and time-to-event data: With applications in R. CRC press.].

Our objective with the current analysis is somewhat more modest, since we wanted to use the longitudinal measurements of CLL prior to study start to predict time to treatment and death. This model enables us to make predictions only from the time

of diagnosis as illustrated in the web application. The clinical problem we had was how could we use the plethora of data available for clinicians at time of diagnosis. We believe that a more advanced approach would also limit the target audience making the study more statistical interesting and less clinical. We have discussed which model to use many times and what time point would be most interesting. We believe that the time of diagnosis serves as a good start, but we agree that a future study could make use of joint modelling making the results more applicable for patients after diagnosis. We are currently working on how to combine (para)clinical variables as employed in CLL-TIM (Agius et al., Nat Comm, 2020) with the approaches employed here to develop a rolling model with prediction of different outcomes during the disease course of CLL.

REVIEWERS' COMMENTS:

Reviewer #2 (Remarks to the Author):

Thank you for making all the changes as requested.

Reviewer #3 (Remarks to the Author):

This manuscript has been great improved. I have no further comments.